# Differences in the Clinical Picture in Women with a Depressive Episode in the Course of Unipolar and Bipolar Disorder

**DOI:** 10.3390/jcm10040676

**Published:** 2021-02-10

**Authors:** Karolina Bilska, Joanna Pawlak, Paweł Kapelski, Beata Narożna, Przemysław Zakowicz, Aleksandra Szczepankiewicz, Maria Skibińska, Monika Dmitrzak-Węglarz

**Affiliations:** 1Department of Psychiatric Genetics, Department of Psychiatry, Poznan University of Medical Sciences, 61-701 Poznań, Poland; jopawlak@ump.edu.pl (J.P.); pkapelski@ump.edu.pl (P.K.); 69633@student.ump.edu.pl (P.Z.); mariaski@ump.edu.pl (M.S.); mweglarz@ump.edu.pl (M.D.-W.); 2Laboratory of Molecular and Cell Biology, Department of Pediatric Pulmonology, Allergy and Clinical Immunology, Poznan University of Medical Sciences, 61-701 Poznań, Poland; bnarozna@ump.edu.pl (B.N.); alszczep@ump.edu.pl (A.S.); 3Center for Child and Adolescent Treatment, 66-003 Zabór, Poland

**Keywords:** mood disorder, coping with stress, impulsiveness scale, stressful life events

## Abstract

Due to current depression prevalence, it is crucial to make the correct diagnosis as soon as possible. The study aimed to identify commonly available, easy to apply, and quick to interpret tools allowing for a differential diagnosis between unipolar and bipolar disorder. The study group includes women with long duration of unipolar (UP, N = 34) and bipolar (BP, N = 43) affective disorder. The diagnosis was established according to the DSM criteria using SCID questionnaire. Additional questionnaires were used to differentiate between UP and BP. BP patients had an earlier age of onset, were hospitalized more times, and more often had a family history of psychiatric disorders than UP (*p*-value < 0.05). Moreover, BP achieved a higher impulsiveness score and more frequently had experienced severe problems with close individuals. To our knowledge, this is the first publication presenting results of numerous questionnaires applied simultaneously in patients on clinical group. Several of them suggest the direction of clinical assessment, such as: the age of onset, family psychiatric burdens, history of stressful life events, learning problems, social and job relations. Further studies are necessary to confirm the utility of this approach.

## 1. Introduction

Depressive episodes are a part of both unipolar (UP) and bipolar (BP) affective disorder. About 4.4% of the global population suffers from UP [1]. Major depressive disorder is a chronic and recurrent disease that manifests with multiple symptoms, including mental and physical features [2]. Major depression significantly impairs an individual’s ability to function at work or school, and cope with everyday life [1]. 

Bipolar disorder has a highly negative impact on a patient’s life. It is evident in many spheres of life, such as relationships with family and friends, employment and difficulty in work, leisure activities, and the quality of life [3,4,5]. A late diagnosis or misdiagnosis is a major challenge in bipolar disorder. Unfortunately, this happens frequently; about 40% [6], or even 70% [7] of patients had a history of previously undiagnosed bipolar disorder. Inappropriate treatments may worsen the disease’s overall course, and may increase mortality due to a high risk for suicide near illness onset [8]. When the proper treatment is initiated near the onset of symptoms, many relapses, and related medical complications, such as substance abuse, are more likely to be avoided [8].

Many authors aimed to find the differences in the characteristic of an episode of depression in unipolar and bipolar disorder, such as an earlier age of onset [9], a more significant number of depressive episodes [10], and greater impact of family history of psychiatric disorders [11,12] in bipolar disorder. Despite these efforts, there are still cases of misdiagnosis that may have irreversible consequences. 

This study aims to identify early markers of the type of depression among biochemical and clinical features readily available to psychiatrists, derived from medical observation and interview. The picture of depression includes both disturbances in the stress axis and disruptions in the circadian rhythm. Therefore, the self-report questionnaires on the chronotype and stress factors preceding the episode were included in the analysis. Compliance with the psychiatrist is essential for controlling the progress of the disease. Thus, the personality traits, especially a high impulsivity level, were also assessed, as it may hinder obtaining therapeutic success. We demonstrated here for the first time the use of numerous questionnaires conducted on one clinical group for better differential diagnosis: The Barratt Impulsiveness Scale, The Coping Orientation to Problems Experienced, Morningness-Evenigness Questionnaires, The Pittsburgh Sleep Quality Index, Epworth Sleepiness Scale, Brief Life Events Questionnaire. Previous studies have used only two or three questionnaires for the same study group. Single tools may not be effective enough or be useful specifically in the sample. The availability of more results for one study group may shed light on the misdiagnosis problem. 

Only women were included in the study to avoid differences between gender, and because female patients are more prone to experience depressive episodes, especially at the onset of affective disorder, and the severity of symptoms requires more frequent hospitalization [13,14,15,16].

## 2. Materials and Methods

### 2.1. Participants

The study group (SG) included 79 women admitted to the hospital due to current depressive episode, aged 18–76 (mean = 42.23 ± 15.06), with a diagnosis of bipolar (N = 43) or unipolar disorder (N = 36) based on DSM-IV criteria. The lifetime diagnosis was established by two psychiatrists based on SCID-I (Structured Clinical Interview for Axis I clinical disorders for DSM-IV) [17,18]. The OPCRIT Checklist [19] was applied to determine the lifetime perspective of major depression and sleep disturbances symptoms. Patients were evaluated twice: upon admission to the hospital in acute state of illness (Pre-treatment) and before discharge, after obtaining improvement (Post-treatment). The severity of depression, mania symptoms, and current functioning was assessed both of these times. The following questionnaires were used (Figure 1): the 17-item version of Hamilton Depression Rating Scale (HAMD) [20], Montgomery–Åsberg Depression Rating Scale (MADRS) [21], Young Mania Rating Scale (YMRS) [22], Antidepressant Side-Effect Checklist (ASEC) [23] and Global Assessment of Functioning Scale (GAF) [17]. The exclusion criteria were YMRS score more than 12 [24], present substance abuse, severe and unstable medical condition, neuropsychiatric illnesses associated with cognitive impairment, or a prior clinical diagnosis of schizophrenia or schizoaffective disorder. Before leaving the hospital, the patients completed a battery of self-questionnaires (Figure 1): Beck Depression Inventory version IA (BDI) [25] in Polish validation [26], The Barratt Impulsiveness Scale version 11 (BIS-11) [27], Morningness-Evenigness Questionnaires (MEQ) [28] with three chronotypes (morning, intermediate, evening) based on Pracki et al. [29], The Pittsburgh Sleep Quality Index (PSQI) [30], Epworth Sleepiness Scale (ESS) [31], The Coping Orientation to Problems Experienced (COPE) [32] in Polish adaptation [33], and Brief Life Events Questionnaire (BLEQ) [34]. All scales used in our study were either in the original form or in the Polish adaptation and were previously validated by their authors. Daily functioning questionnaires were filled in in the post-treatment state, since during exacerbation the patient’s judgment is biased towards depressive beliefs and might not reflect the real parameters. The discrepancies in the numbers of participants in several analyses were due to not fully or properly completed questionnaires. Patients were recruited between November 2017 and March 2019 in the Department of Psychiatry in-patient clinic, University of Medical Sciences in Poznan. Pharmacological therapy was administered to all patients included. The data on possible psychological interventions were not available.

The control group (CG) included 69 healthy women volunteers, aged 22–63 (mean = 42.54 ± 11.25). CG completed the same five self-questionnaires as SG (ESS, MEQ, PSQI, BDI, BLEQ) to facilitate the recruitment. The exclusion criteria were: more than 10 points obtained in BDI, more than 14 in ESS, shifts workers, and a positive personal history of psychiatric symptoms. 

Blood samples were collected from the control group and patients (in pre- and post-treatment state), after which basic blood tests (blood count and biochemical analysis) were performed.

The study was approved by the Bioethics Committee of Poznan University Medical Sciences (decision 758/17 22 June 2017). All study participants were Caucasians of Polish origin and gave written informed consent.

### 2.2. Statistical Analyses

The distribution of variables was studied by the Shapiro–Wilk test. Variables with normal distribution were tested using parametric tests: student’s t-test for comparing two groups and ANOVA for several groups. If the results have shown statistically significant differences, Tukey’s post-hock test was used. For variables that did not meet the criteria of a normal distribution for statistical comparisons, non-parametric tests were used between the examined groups: Mann–Whitney U test distributions for two independent groups, Wilcoxon pair order test for two dependent groups, and ANOVA rank Kruskal–Wallis test for several independent groups. Discrete measures were assessed with the chi-square test. To assess the relationship between the analyzed variables, Spearman’s rank correlation coefficient was applied. Two-sided comparison of two means test was used to compare literature data with our results (mean, standard deviation, and group size were known). The significance level was set at *p* < 0.05 for all analyses. Statistical calculations were made using the STATISTICA 13.3 (StatSoft, Krakow, Poland).

## 3. Results

### 3.1. Sample Characteristics Including Social Status and Basic Clinical Data

Several sociodemographic data, including age, education years, and marital status, were collected from the study and control group (Table 1). There was no significant difference between groups in terms of size and age, whereas average years of education were significantly higher for the control group (CG). The comparison across marital status showed that the study group (SG) is more likely to be divorced than CG (Table 1). Moreover, we have observed fewer married persons in SG (41.10%) compared to CG (68.66%). Between disorder groups, BP patients (32.50%) were less likely to be married than UP patients (51.51%), but these differences were not significant. In SG, 64% of participants who declared their current occupation as “retired” did not meet the age criterion (above 60 years of age) but had health pension due to a mental disorder.

Patient’s medical history data, such as the age of onset, duration of hospitalization in the present episode, and the number of hospitalization, were compared between the UP and BP group (Table 1). Patients with bipolar disorder had an earlier age of onset and Not all participants had all of the data collected, therefore the size of the group may differ in the individual items. were more frequently hospitalized than unipolar patients (correlation coefficient 0.4077). In addition, age of onset showed weak negative correlation with marital status in BP patients (correlation coefficient −0.3297) and strong in UP patients (correlation coefficient −0.7110). This means that the earlier age of onset, the more people are unmarried in both diagnoses. Moreover, bipolar patients (78.05%) reported more frequently a family burden of psychiatric disorders in first and second-degree relatives than unipolar patients (54.55%). In the present episode, BP patients were hospitalized longer (7.2 ± 3.68) than UP patients (5.7 ± 2.18), however this difference was not significant. Spearmen’s rank correlation showed positive correlation between long depressive episode in the past and long hospitalization in the present episode in BP patients (correlation coefficient 0.3677). Interestingly, UP patients showed positive correlation between age and hospitalization time (correlation coefficient 0.4675); older persons were hospitalized longer. We have also compared the history of suicide attempts and substance use disorders between these two groups, but there were no significant differences.

### 3.2. Depressive Symptoms Severity in the Pre- and Post-Treatment State

Hamilton’s total score was higher in pre-treatment patients compared to post-treatment ones, in both diagnoses (Table 2). Seven patients (2 BP and 5 UP) did not achieve remission (score ≤ 7) [35], but they showed improvement >50% in the total score. MADRS total score was also higher in pre-treatment patients than post-treatment ones, in both diagnoses. Only two patients (2 BP, different ones than in HAMD) did not achiever emission (score < 10) [36], but they had improvement >50% in the total score. The current mental state was also assessed by applying self-questionnaire BDI (Table 2). This tool allows comparison of patients groups with controls. All differences between pre-treatment, post-treatment, and controls were significant (Kruskal–Wallis test, post-hoc Dunn test, *p*- and *z*-value < 0.05). There were no statistically significant differences in the following scales: HAMD, MADRS, YMRS, and BDI, between the diagnostic UP vs. BP group in pre- and post-treatment assessment. Based on SCID-I we compared depressive symptoms profile between the diagnoses (Appendix A). Three symptoms were reported by BP patients more frequently (chi-square *p* < 0.05). These were hypersomnia (0.00% UP and 17.95% BP), suicidal ideation (50.00% UP and 76.92% BP) and specific suicidal plan (33.33% UP and 53.85% BP). Other symptoms did not differ between the diagnoses, although suicide attempt was near significant higher (*p*-value 0.0555) in BP.

### 3.3. Analysis the Group of Medications Taken in the Pre- and Post-Treatment State

We have collected a list of drugs that were administered to each patient upon admission to the hospital and during the hospital stay. The number of medicines taken was compared between the diagnoses, but there were no statistical differences. However, the number of drugs taken was higher in post-treatment patients than in the pre-treatment group (Table 2). The most frequently used medicines belong to five psychiatric drug groups: selective serotonin reuptake inhibitor, selective noradrenalin-serotonin reuptake inhibitor, mood stabilizers, benzodiazepines, and neuroleptics/antipsychotics. Among mood stabilizers, lithium carbonate was prescribed in monotherapy or in combination with another mood stabilizer for 18.60% BP subjects and 2.94% UP subject in the pre-treatment stage, while 25.58% of BP subjects and 5.88% UP patients were prescribed other mood stabilizers (e.g., carbamazepine) either alone or in combination. In the post-treatment assessment, utilization of mood stabilizers increased. Lithium was taken as frequently as other mood stabilizers in BP patients (34.88%). In the UP group, lithium was prescribed to 14.71% of subjects, and other mood stabilizers were taken by 8.82% of individuals, although the diagnosis was not changed. As expected, bipolar patients were more likely to have prescribed mood stabilizer and neuroleptics/antipsychotics than patients with major depressive disorder (U-Mann–Whitney test, *p* < 0.05). 

### 3.4. Assessment of Global Functioning in the Pre- and Post-Treatment State

Current functioning was measured with the Global Assessment of Functioning scale [17] in pre- and post-treatment states (Table 2). There were significant differences between states in both diagnoses, as we observed improved functioning of the patients. However, there were no differences between BP and UP groups both in pre- and post-treatment assessment. They have shown a similar pattern in functioning. In the pre-treatment state, patients were usually classified to score bracket 41–50, whereas in the post-treatment state were observed improvement to category 71–80.

### 3.5. Laboratory Findings

Pre-treatment patients in both diagnoses had a higher CRP score, triglyceride, and LDL than controls, whereas HDL and TSH had lower value in the study group (Appendix A). It is worth to pointing out that the average CRP, total cholesterol and HDL values exceeded the acceptable reference standards in the pre-treatment state (Appendix A). Within the pre- and post-treatment state only the TSH level differed significantly in patients with bipolar disorder. The mean value was higher (Wilcoxon test, *p* = 0.0268) in post-treatment (3.02 µlU/mL) than the pre-treatment state (1.74 µlU/mL).

### 3.6. Sleep Quality and Chronotype Analysis in BP, UP and CG

Several of the clinical questionnaires (HAMD, MADRS, BDI) inquire whether the patients have sleep problems (sleep disturbance, light sleep, intermittent, early awakening, shortening of sleep). The analysis of the results obtained in the pre- and post-treatment state showed improved sleep quality in all analyzed items (Wilcoxon test, *p* < 0.05). The results were compared between the diagnoses; higher values for sleep disturbance (HAMD question 4) and sleep deterioration (BDI question 16) were observed in UP than in BP (Mann–Whitney U test, *p* < 0.05). The results indicate significant differences in the frequency of chronotypes (Appendix A). The most often observed type in SG was the evening chronotype, whereas the least often observed was the morning chronotype (Chi-square test, *p* = 0.0038). In CG, intermediate chronotype was the most often found, and, similar to SG, the least observed was morning chronotype (Chi-square test, *p* = 0.0324). In comparison between patients and the control group, we discovered a difference in the frequency of choosing intermediate chronotype (Mann–Whitney U test, *p* = 0.0194) (Appendix A). Moreover, the frequency of chronotype for UP differed significantly from those of CG (Kruskal–Wallis test, post-hoc Dunn test *p* = 0.0489 and *z* = 2.4024) and showed a pattern represented by the whole SG, whereas BP does not vary considerably from CG.

From the questionnaires concerning the quality of sleep and sleepiness (PSQI and ESS), we have chosen several sleep variables and compared them between controls and subgroups of patients (Table 3). Three characteristics (sleep latency, medicine-induced sleep, PSQI) were significantly worse in patients than in controls, whereas sleep duration was shorter in the control group. 

The ESS results (daytime sleepiness) were similar for BP and control groups, but differed significantly between UP and controls (Table 3). UP achieved lower scores in ESS. Scores above 14 were achieved by 11.11% of BP group and 8.33% of UP patients. During the comparison of ESS total score with chronotype, we have observed a statistically significant correlation (*p* = 0.0191).

Patients presenting the evening chronotype were more likely to have high ESS scores. There were no differences in sleep variables (Table 3) when compared to BP and UP groups. Sleep problems during illness episodes are described in several items of OPCRIT, and were correlated with chronotype, PSQI total score, ESS and questions from HAMD and BDI concerning sleep problems (studied only in SG). Significant positive correlation (Spearman’s rank correlation coefficient *p* < 0.05) was observed between OPCRIT and questionnaires variables (Table 4). As is shown in the Table 4, there are differences in the number and type of thematic issues that correlate in BP (4) and UP (13) patients. However, three of the four issues relevant to BP coincide with those for UP. Higher number of issues with positive correlations between the questionnaires for UP patients, may indicate a greater severity of sleep problems in these patients.

### 3.7. Impulsivity Assessment in BP and UP

The Barrat Impulsiveness scale scores are presented in Appendix A. The average total BIS-11 score for the entire study group was (62.7 ± 7.1), with the lowest (39) and highest (75) values outside the normal range (52–71), as proposed by Stanford et al. [37]. In our study, there were two individuals (UP) with a lower score than the normal range and three (BP) with higher results. UP scored significantly lower than BP groups (Mann–Whitney U test, *p* = 0.0152). First- and second-order factor level differences were shown on Motor impulsiveness; BP scored substantially higher than UP. The impulsive patients are more likely to suffer from bipolar disorder. Moreover, total BIS-11 score showed positive correlation with the item “medicine-induced sleep” (correlation coefficient 0.5459) and PSQI total score (correlation coefficient 0.6040) in BP. There was no such relationship for UP patients.

### 3.8. Stressful Life Events before the Relapse

Based on BLEQ, patients experienced a greater number of stressful life events in comparison to the control group (Appendix A). In SG, 73.58% of patients reported one or more stressful situations, whereas, in CG, only 48.53% did. Participants with a mood disorder more commonly experienced five kinds of events than controls. These events were related to severe illness (event 1), death of a spouse or first-degree relative (event 3), separation due to marital difficulties (event 5), a serious problem with a close friend (event 6), and a job loss (event 7). Two kinds of events differed between diagnoses: UP experienced more often the death of a close ones, whereas BP pointed out problems with friends.

### 3.9. Coping with Stress Assessment in BP and UP in Comparison to Literature CG

COPE questionnaire differentiates 15 strategies for coping with stress. Alcohol/Drug Use and Humor were the least frequently reported by patients with affective disorder (Appendix A). The obtained results did not differ significantly between the diagnoses. The clinical group was compared to the literature data by Juczyński and Ogińska-Bulik [38]. The following significant differences were found (see Appendix A): our patients chose less frequently Seeking Instrumental Social Support and Positive Reinterpretation and Growth strategies. They have also more often selected Behavioral Disengagement than the controls.

### 3.10. Antidepressant Side-Effect Before and After Treatment

Participants reported mild side effects, except for one person, who reported moderate symptoms in the pre-treatment state (insomnia, headache, and orthostatic dizziness) and post-treatment state (appetite increase). About half of the symptoms differed significantly between pre- and post-treatment state in bipolar disorder (Appendix A). In most cases, the severity of these symptoms was lower after hospital treatment than in the beginning, except for the increased appetite. In unipolar disorder, only two symptoms differed significantly (insomnia and decreased appetite). Similarly to BP, decreased in the post-treatment stage. The only difference between the diagnoses was reported in the pre-treatment state concerning the dry mouth; BP patients described it more often. In both diagnoses, patients reported fewer complaints in post-treatment than pre-treatment state (Wilcoxon test, *p* = 0.0038 for BP and *p* = 0.0074 for UP). Although experienced psychiatrist conducted the medical interviews the results should be treated with some caution. It is challenging to distinguish the side effects of drugs from the disease symptoms, especially in patients that had the disease for many years and take several different medications. Moreover, depressed patients might be more prone to report side effects due to an overall more negative assessment of reality.

## 4. Discussion

Our study showed that BP patients had an earlier age of onset, had more frequently a family burden of psychiatric disorders, and they required hospitalization more often than UP patients. Moreover, BP patients were more impulsive and more frequently had a suicidal ideation and specific suicidal plan than UP patients. These data were spread in numerous clinical assessment tools. This suggest that not the specific tool is crucial for differential diagnosis, but rather an ensemble of factors investigated. 

### 4.1. BP Patients Struggle More with Social Relationships Than UP Subjects

It is crucial that the differential diagnosis between UP and BP depressive episodes is made as soon as possible. Bipolar disorder was reported to have a negative impact on everyday life, including the ability to take care of daily tasks, difficulties in the performance of work-related activities, social adaptation, and problems with relationships (conflicts within families and friends, marital difficulties, and employment), as well as the ability to manage financial responsibilities [3,4,5]. Bipolar patients experience extreme and sudden mood swings, which often strain relationships, ruin friendships, and destroy careers. Cooperation with that person can be highly frustrating and could be a source of misunderstanding and confrontation. People with bipolar disorder are more to get divorced or never to get married at all [39]. In our study, bipolar patients who were divorced or never married accounted for over 62% of all BP. For comparison, this was about 30% and 40% in controls and unipolar patients, respectively. It clearly indicated that BP relationships are more likely to fail, whereas UP did not differ significantly from controls.

Another challenge for people with bipolar disorder is the problem of keeping a job. In our research, 75% of bipolar patients have had a job before the onset of disease; later on, only 30% were employed. Bipolar patients handle worse compared to unipolar patients (46.87% have a job). More than half of retired patients belong to this group because of the severity of the mental disorder (not because of age criteria). Morseli et al. [40] made similar observations for bipolar disorder. In addition, bipolar patients were more frequently hospitalized than unipolar patients [41]. In our study, bipolar patients were two times more often hospitalized than unipolar patients. Bipolar patients showed a lower quality of life in social functioning, performing daily activities, working and accomplishing tasks compared to unipolar patients [4,42,43]. Patients with bipolar disorder also had lower educational attainment than controls [44]. In our study, BP and UP patients had fewer years of education than controls. Educational failure may contribute to later long-term impairment in occupational and social functioning [44].

### 4.2. Evening Chronotype and Daytime Difficulties Are Common 

Our patients predominantly showed evening chronotype, which is consistent with the previous studies [2,45]. This chronotype shows a correlation with the higher ESS results, which indicate daytime sleepiness. Mume [46] measured daytime sleepiness during a depressive episode. The mean ESS score in that study was above 9 (in our post-treatment patients was near 7), and the researcher observed a high, positive correlation between excessive daytime sleepiness and severity of depression. Johns [31] demonstrated that a mean score around 6 is characteristic for healthy controls. It clearly shows that the illness has an impact on daytime sleepiness. Ozcelik and Sahbaz [2] used the Biological Rhythms Interview of Assessment in Neuropsychiatry (BRIAN). They revealed not only problems with sleep, but also disturbances in social relations, activity, and eating. The study group showed longer sleep latency and higher sleep-inducing medicine use than controls. Similar results were achieved in an earlier study [45]. In the control group, we observed a shorter sleep duration than in the study group. This might result from the fact that control subjects have more duties related to work and family, and need to get up earlier. It may be also due to the effect of sleep medications taken by the SG.

### 4.3. BP Patients Are More Impulsive Than UP Subjects

Impulsiveness is defined as a predisposition to act rapidly without reflection and without assessing such behavior’s negative consequences [37]. Impulsive action is related to several socially deviant behavior such as aggression or substance abuse, but also some psychiatric disorders, such as borderline personality, kleptomania [17,37] or bipolar disorder [47]. For individuals with bipolar disorder, impulsivity contributes to a suicide attempt risk [48] and comorbidity with substance use disorders [49]. Patients who have attempted suicide in the past tend to score higher on the BIS-11 [37]. Swann and co-workers [50] report that BIS scores were increased in euthymic bipolar patients with present or past substance abuse. Our study did not reveal a higher total BIS 11 score in suicide attempters, but these study group was small. The only differences were found in slightly higher Motor Impulsiveness for suicide attempters (21.0 ± 3.6) than non-attempters (20.0 ± 3.2). A similar pattern was shown in the case of patients with and without substance use disorders. All these results were not statistically significant. 

Euthymic unipolar [47,51] and euthymic bipolar patients [47,50,51] achieve higher impulsiveness scores than healthy individuals. According to the study [50], manic and euthymic patients in the course of BP achieved similar levels of trait impulsivity, while Peluso et al. [47] found that depressed and euthymic bipolar patients exhibit similar levels of this trait. Only euthymic UP scored significantly lower than depressed UP and BP [47]. In our study, the post-treatment UP achieved a lower score than post-treatment BP. This indicates that even in euthymia, BP patients are more impulsive than UP, which may be reflected in more frequent relapses and suicide attempts. In our study, it was observed as a higher number of hospitalizations in BP (5.60 ± 7.57) than in UP (2.00 ± 2.26). 

### 4.4. Relationship Problems Are More Likely to Trigger Episode in BP, Whereas UP Patients More Often Have a Relapse after Bereavement 

In this work, the history of adverse life experiences: personal illness, close family death, interpersonal problems, and job loss, were associated with mood disorders. Hosang et al. [52] showed that the majority of stressful life events linked with unipolar depression are also associated with bipolar disorder. However, several events were found to be more pertinent in unipolar than bipolar disorder, and conversely. The death of a spouse or a first-degree relative was more strongly related to the unipolar disorder, whereas a serious problem with a friend, neighbor, or relative was related to bipolar disorder [52]. These observations are consistent with our results: UP patients more prone to react with symptoms in bereavement, while BP depression is more often linked to relationship security.

### 4.5. Affective Disorder Patients Use Behavioral Disengagement Strategy

Our results of the COPE inventory did not differ between the diagnoses. A similar result was found by Engel-Yeger and colleagues [53], whereas Coulston et al. [54] reported that the BP group had a significantly higher score than UP on adaptive coping. This category includes active coping, the use of instrumental social support, planning, and positive reinterpretation. That study found no differences between the BD-I and BD-II groups [54]. The study mentioned above used the same version of the COPE inventory that we used. In the literature there are examples of several differences between the type of BP, but other researchers used Brief-COPE questionnaire [55], which cannot be directly compared. Fletcher et al. [55] showed that bipolar II patients were less willing to seek support in stressful situations and less likely to engage in down-regulated hypomania strategies. In our study, the post-treatment patients differed from the controls. Patients more often chose Behavioral Disengagement. Even in euthymic state, patients are more likely to give up trying to reach their goals or solving the problems than control. Unfortunately, the consequences of such behavior can have a serious impact on occupational work as well as personal life. In our study, it was visible in higher unemployment in SG than CG, and also in more frequently reported social relationships problem, especially in BP.

## 5. Conclusions

It is crucial to distinguish whether the depressive episode occurs in the course of UP or BP. The profile of clinical symptoms does not allow to differentiate it during the first episode. Therefore, a detailed medical interview with the patient is essential to determine the time of the onset of the disease, family burdens, stressful life situations, difficulties in social relations, suicidal thoughts, and the ability to keep a job. OPCRIT is a tool that collects all of the above-mentioned variables in one place. However, several questionnaires, which are more detailed, might be helpful in clarifying the severity of symptoms. One of them is the BDI. This utility is used to assess the severity of the depression by inquiring about other related areas, such as suicidal thoughts, sleep, work, appetite, and interest in other people. Since the test has structured answers instead of “yes” or “no” responses, it is possible to evaluate the severity of each of the symptoms. For example, inquiry about suicidal thoughts shows whether the patient is only thinking about it but is unable to attempt suicide, or if he has a plan and is waiting for the right circumstances. Such an approach allows us to select important factors that could be emphasized in order to improve the patient’s condition. Based on this questionnaire, in our study, unipolar patients more often reported sleep deterioration than BP patients. HAMD is a similar, but more objective questionnaire, as it is filled in by the psychiatrist and not by the patient, as in the case of the BDI. Similarly to BDI, HAMD also showed sleep disturbance more often in UP than in BP. Data on sleep patterns, taken from symptom profiles or additional questionnaires, could be beneficial for differentiation diagnosis and treatment. In our study, UP was less similar to controls in terms of chronotypes (MEQ questionnaire) and ESS. In contrast, hypersomnia (OPCRIT) is more likely to be found in BP patients. Another important aspect is the stress situation before relapse. The most frequently stressful situations and their impact on patient’s psychical health are listed in BLEQ, an easy-to-use tool that allows to find out which stressful situations might have caused the relapse of the disease. Our results indicated that BP patients had more often problems with friends (with social relationships in general), whereas UP patients more frequently experienced death of a close ones.

Impulsiveness is an important personality trait in the context of attempting suicide. It has been shown that more impulsive people make such attempts more often. BIS-11 is a proven test that enables the assessment of impulsivity. Our results had shown that impulsive patients are more likely to suffer from bipolar disorder.

Our research shows that BP patients are more often unemployed than UP. During the interview with the patient, the doctor should pay attention whether the patient changes jobs frequently or remains unemployed for a long time, as this could help with the correct diagnosis.

There are various advantages and disadvantages to those methods; however, the latter is due to the fact that self-assessment tests are burdened with the patient’s subjectivity, and not all language versions are validated for psychiatric use. Our study demonstrates the legitimacy of using the additional tools among available questionnaires. In our opinion, additional questionnaires, especially those focusing on the age of onset, family predispositions, suicide attempts, sleep problems, impulsiveness, and stressful situations before relapse, could help with the differential diagnosis and correct treatment. We have tested only several tools in this study, but we believe that other tests covering these subject areas may also be useful. Considering numerous aspects and questionnaires, the risk of misdiagnosis might be reduced with the help of several tools. Thus, treatment strategy might be better individualized, contributing to a better quality of life for the patient, fewer relapses and a more favorable course of a disorder.

## 6. Limitations

The main limitation of this study is a small research group that completed the questionnaire correctly. Our study group included only women because of the differences in the course of disease between genders. Men with bipolar disorder are more prone to substance use disorders [13,56,57]. In our study, we included only the subjects without comorbidity with substance use disorder. We did not compare BP I vs. BP II patients because the number of participants for analysis was too low.

## Figures and Tables

**Figure 1 jcm-10-00676-f001:**
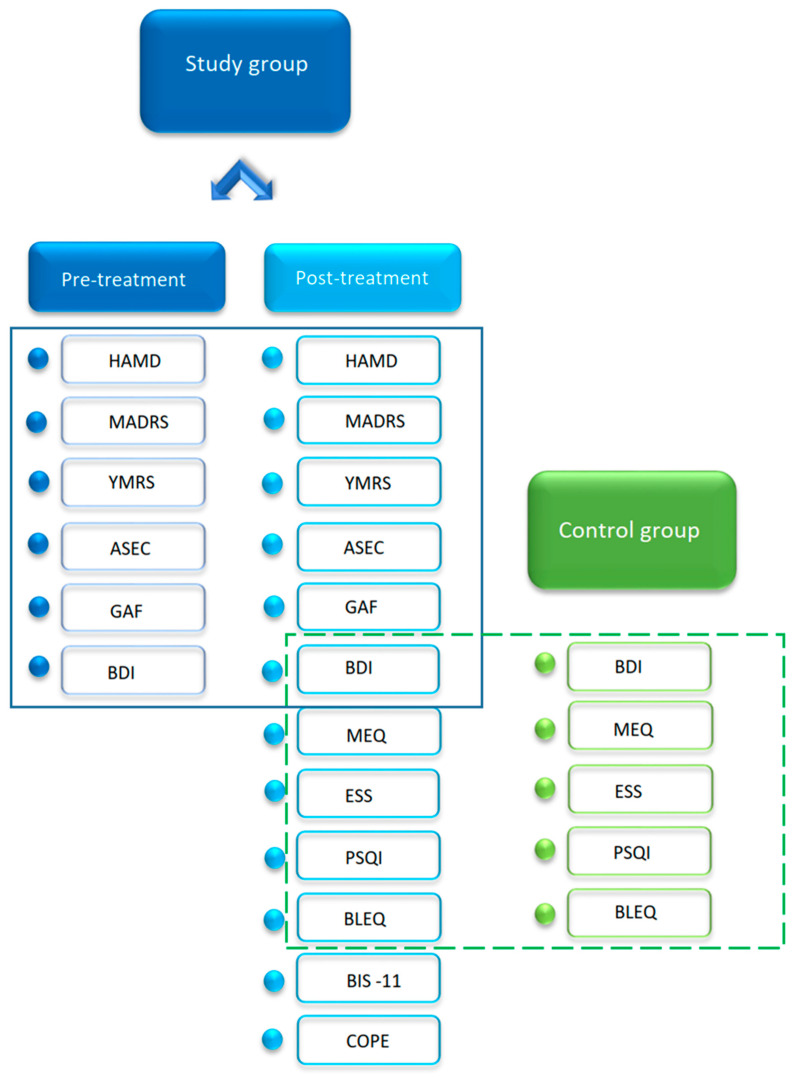
Questionnaires used in the study. The box marks the common tests for each group: the solid line for patients in pre- and post-treatment state, and the dotted line for post-treatment patients and control group. Abbreviations: HAMD—Hamilton Depression Rating Scale, MADRS—Montgomery–Åsberg Depression Rating Scale, YMRS—Young Mania Rating Scale, ASEC—Antidepressant Side-Effect Checklist, GAF—Global Assessment of Functioning Scale, BDI—Beck Depression Inventory version IA, MEQ—Morningness-Evenigness Questionnaires, ESS—Epworth Sleepiness Scale, PSQI—The Pittsburgh Sleep Quality Index, BLEQ—Brief Life Events Questionnaire, BIS-11—The Barratt Impulsiveness Scale version 11, COPE—The Coping Orientation to Problems Experienced.

**Table 1 jcm-10-00676-t001:** Sociodemographic data.

Characteristics	Study Group	Control Group	*p*-Value	BP	UP	*p*-Value
Female, N	79	67	0.3206 ^ƚ^	43	34	0.3051 ^ƚ^
Age, years: mean [SD]	42.23 [15.06]	42.46 [11.41]	0.8599	41.33 [14.24]	43.31 [16.12]	0.6226
Education, years: mean [SD]	13.65 [2.78]	15.81 [2.15]	<0.0001	13.39 [2.84]	13.97 [2.71]	0.2969
Marital status, N [%]						
single	25 [34.24]	17 [25.37]	0.2170 ^ƚ^	15 [37.50]	10 [30.30]	0.3173 ^ƚ^
married	30 [41.10]	47 [68.66]	0.0527 ^ƚ^	13 [32.50]	17 [51.51]	0.4652 ^ƚ^
divorced	13 [17.81]	3 [4.48]	0.0124 ^ƚ^	10 [25.00]	3 [9.09]	0.0522 ^ƚ^
widow	5 [6.85]	1 [1.49]	0.1025 ^ƚ^	2 [5.00]	3 [9.09]	0.6547 ^ƚ^
Occupation before disorder, N [%]						
employed				30 [75.00]	20 [64.52]	0.1573 ^ƚ^
unemployed				10 [25.00]	11 [35.48]	0.8273 ^ƚ^
Current occupation, N [%]						
employed (full or part-time work)	27 [37.50]	65 [97.01]	<0.0001 ^ƚ^	12 [30.00]	15 [46.87]	0.5637 ^ƚ^
student	11 [15.28]	0 [0]	0.0009 ^ƚ^	7 [17.50]	4 [12.50]	0.3657 ^ƚ^
unemployed or disable (including ‘retired’)	34 [47.22]	2 [2.99]	0.0001 ^ƚ^	21 [52.50]	13 [40.63]	0.1701 ^ƚ^
Family history of psychiatric disorders, N [%]	50 [67.57]	0 [0]	<0.0001 ^ƚ^	32 [78.05]	18 [54.55]	0.0477 ^ƚ^
Age of onset: mean [SD])				28 [9.60]	35 [14.8]	0.0295
Duration of hospitalization, weeks: mean [SD]				7.20 [3.68]	5.7 [2.18]	0.1603
Number of hospitalizations: mean [SD]				5.60 [7.57]	2.00 [2.26]	0.0004

BP—bipolar disorder, UP—unipolar disorder, ^ƚ^–chi square test, Mann–Whitney U test was used for other data, significant *p*-value in bold.

**Table 2 jcm-10-00676-t002:** Questionnaire total score for the study group.

Measure	Bipolar Disorder					Unipolar Disorder				
	Pre-Treatment		Post-Treatment		*p*-Value	Pre-Treatment		Post-Treatment		*p*-Value
	Mean	SD	Mean	SD	Wilcoxon Test	Mean	SD	Mean	SD	Wilcoxon Test
Hamilton Rating Scale for Depression–17	26.03	8.23	3.39	2.42	<0.0001	25.51	6.10	3.85	3.09	<0.0001
Montgomery-Asberg Depression Rating Scale	28.53	6.37	4.82	4.08	0.0003	31.80	6.73	3.80	3.16	0.0051
Young Mania Rating Scale	1.39	1.86	0.29	0.90	0.0029	0.93	1.62	0.03	0.19	0.0077
Global Assessment of Functioning scale	46.64	9.72	76.50	9.32	0.0010	46.89	14.12	79.00	4.90	0.0077
Number of psychotropic medications taken	4.45	2.05	5.39	2.91	0.0058	3.32	2.15	4.71	2.97	0.0019
Beck Depression Inventory	35.39	8.64	8.82	8.56	<0.0001	31.04	13.41	10.73	10.35	<0.0001

Beck Depression Inventory total score in control group 4.54 ± 3.47; *p*-value in bold are statistically significant.

**Table 3 jcm-10-00676-t003:** The Pitsburgh Sleep Quality Index sleep variables differences between controls and selected groups of patients.

	Compared Groups	Control Group		Bipolar Patients		Unipolar Patients		*p*-Value of Kruskal–Wallis Test	Post-Hoc Dunn Test *p*- and *z*-Value		
Variables		Mean	SD	Mean	SD	Mean	SD		CG vs. BP	CG vs. UP	BP vs. UP
Sleep duration		0.6269	0.8499	0.2222	0.5774	0.1481	0.6015	0.0008	0.1059	0.0244	1.0000
									(2.1050)	(2.6470)	(0.4539)
Sleep disturbances		0.8955	0.4650	1.0769	0.3922	1.0370	0.1925	0.0899	0.6338	0.9445	1.0000
									(1.2501)	(1.0051)	(0.2173)
Sleep latency		0.7761	0.7349	1.3333	0.8771	1.7037	0.7753	0.0000	0.0246	0.0000	0.4468
									(2.6437)	(4.3670)	(1.4433)
Daytime dysfunction		1.0000	0.5505	1.2400	1.0520	1.2963	0.8234	0.2492	1.0000	0.4527	1.0000
									(0.7583)	(1.4364)	(0.5394)
Habitual sleep efficiency		0.3582	0.6898	0.4074	0.7971	0.6296	0.8389	0.1355	1.0000	0.3454	0.6475
									(0.0977)	(1.5755)	(1.2377)
Sleep quality		0.8955	0.6064	0.7778	0.6405	1.0000	0.6794	0.4717	1.0000	1.0000	0.9061
									(0.7081)	(0.5241)	(1.0321)
Medicine-induced sleep		0.2388	0.7196	1.0370	1.1923	1.5926	1.4212	0.0000	0.0134	0.0002	1.0000
									(2.8422)	(3.9887)	(0.9603)
PSQI		4.7910	2.5198	6.1600	3.6019	7.4074	3.0415	0.0007	0.3160	0.0005	0.2626
									(1.6195)	(3.7453)	(1.7085)
Chronotyp		1.9104	0.7330	1.6667	0.7338	1.4800	0.7141	0.0255	0.5132	0.0489	1.0000
									(1.3687)	(2.4024)	(0.9044)
ESS		7.5970	3.0254	7.5556	4.2547	5.9167	4.4027	0.0890	1.0000	0.0941	0.2868
									(0.1951)	(2.1523)	(1.6665)

CG—control group, BP—bipolar unipolar, UP—unipolar disorder, PSQI—global score of sleep quality, ESS—global Epworth Sleepiness Scale. Post-hoc Dunn test *p*- and *z*-value are given in one row; *z*-value in brackets. Significant *p*- and *z*-value (*p* < 0.05) in bold.

**Table 4 jcm-10-00676-t004:** Correlation of sleep issues between different questionnaire for BP and UP patients.

BP	Type of Questionnaire and Thematic Issues	HAMD		BDI	PSQI			
		Intermittent, Shallow Sleep	Premature Waking Up	Premature Waking Up	Sleep Disturbances	Medicine Induced Sleep	Daytime Dysfunction	Total Score
OPCRIT	reduced need for sleep	−0.2877	−0.0376	0.3238	0.0751	0.1119	−0.1507	0.1268
	difficulty falling asleep	0.4373	−0.0234	−0.0698	−0.0652	0.1830	−0.2387	−0.0291
	waking up at night	0.4227	0.0384	0.0499	−0.0602	0.0000	−0.2169	−0.2285
	premature waking up	0.2486	0.5411	0.4242	0.1780	0.0338	−0.3745	−0.0145
	excessive sleepiness	−0.2509	−0.2982	−0.1462	−0.2221	−0.0192	0.4026	−0.0702
**UP**	**Type of Questionnaire and Thematic Issues**	**HAMD**		**BDI**	**PSQI**			
		**Intermittent, Shallow Sleep**	**Premature Waking Up**	**Premature Waking Up**	**Sleep Disturbances**	**Medicine Induced Sleep**	**Daytime Dysfunction**	**Total Score**
OPCRIT	reduced need for sleep	0.1062	−0.0262	0.0689	0.4703	−0.1530	0.0072	−0.1819
	difficulty falling asleep	0.3887	0.1809	0.4259	0.1504	0.1363	−0.2852	0.1197
	waking up at night	0.1916	0.1681	0.5965	0.0993	0.3362	−0.1123	0.2333
	premature waking up	0.1648	0.5663	0.4524	0.2296	0.6643	−0.2935	0.4936
	excessive sleepiness	−0.1419	0.0090	−0.0235	0.1248	0.3317	0.3850	0.1753

The top part of the table shows the results for BP patients, whereas in the bottom part are results for UP patients. Spearman’s rank correlation, significant coefficient (*p* < 0.05) are in bold. Abbreviations: HAMD—Hamilton Depression Rating Scale, BDI—Beck Depression Inventory, PSQI—Pittsburg Sleep Quality Index.

## Data Availability

The data presented in this study are available on request from the corresponding author. The data are not publicly available due to study is ongoing.

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
