# Peer review of "Differences in the Clinical Picture in Women with a Depressive Episode in the Course of Unipolar and Bipolar Disorder"

_jcm, 2021, doi:10.3390/jcm10040676_

Round 1

Reviewer 1 Report

In the manuscript entitled “Differences in the clinical picture in women with a depressive episode in the course of unipolar and bipolar disorder”, Bilska and colleagues aimed to identify adequate diagnostic tools for a an early differential diagnosis between patients with unipolar (UP) and bipolar (BP) disorder.

  1. This is a complex study using multiple diagnostic tests. The study, in my view, has been carefully designed and analysed. However, it is not entirely clear what it the novelty in the authors’ approach, except using many tests in UP and BP women patients in pre-treatment and post-treatment stages. Reading the discussion the overall impression is that most of the observations have already been made and published. The authors should highlight the findings of their work.
  2. Additionally, what I am missing in the manuscript is a broader reflection about utility of these tests (used together as the authors suggest or only some of them) in the clinical practice and conclusions or at least suggestions which from the studied parameters (maybe a few or all that statistically differ between the groups?) are the ones that should be tested/checked in order to (at least with the higher probability) diagnose a patient in the early stage of disease, and to distinguish between UP and BP, what will allow to implement the appropriate treatments and, possibly, improve the overall quality of life in both patients’ groups.
  3. As the authors admit, due to a relatively small number of participants this is a pilot study, but some suggestions regarding the future study approach should be made.
  4. In some parts of the DISCUSSION, the interpretation of the results is missing e.g. P10L110-111, P11L161, P12L186-187

Minor comments:

  • P2L84-85: Information during what period patients were recruited to the study should be provided. How long they were hospitalized? Where there any differences between UP and BP patients related to the hospital stay?
  • In the RESULTS or DISCUSSION, the titles of the subsections should be in a form of a conclusion/summary of the obtained results and not repeated. This would make much easier for the reader to “get the message” of a paper.
  • The good practice is also providing, at the beginning of the DISCUSSION, a few short sentences describing the main finding/s of the study…
  • Explanation of the used abbreviations should be provided in a legend for Figure1 (P3) and in description for Table S3
  • Small language, punctuation, and typo corrections are required e.g. P6 sentence before 3.3 subsection; P9L67, P10L114, P12L213, P18L294, P18L306 (“statistically significant” not important), description for Table S1, point 12 in Table S7

Author Response

Dear reviewer,

We appreciate the time you took to review our manuscript. Thank you very much for your helpful comments and corrections. We tried to edit the manuscript accordingly. All changes are highlighted in yellow. The detailed answers to your question are in the attachment file.

Best regards,

Karolina Bilska  

Reviewer 2 Report

The current study investigated a plethora of variables (e.g., age of onset, family history of psychiatric disorders) that might serve as early markers for a quick differentiation between unipolar and bipolar patients. Results showed individuals with BP have an earlier onset and are hospitalized more times than individuals with UP. There are some major issues that need to be addressed before I can recommend the manuscript for publication.

- As stated above, a plethora of comparisons was carried out without correcting for false positives. Some form of correction method (e.g., Benjamini-Hochberg) should be used.

- I was wondering why post-treatment scores are shown. There´s no rationale in the introduction nor results are discussed. It remains unclear what type of treatment was administered (e.g., cognitive behavioral therapy) and whether both subgroups were comparable in type and duration of treatment. Did one group benefit more from the treatment? This could be checked with an ANCOVA with pretest scores as a covariate.

- In this vein, UP and BP were compared separately for pre and post-scores (e.g. global functioning). However, both groups were not compared directly with each other pre-treatment. If conclusions about early markers are drawn, this would be an important contrast.

- The manuscript would profit from an additional section describing the scales in more detail (example items; Scaling of the rating scales; Cronbach´s alpha in the current sample). Otherwise, it´s unclear if questionnaires can be used in the current sample (e.g. due to low reliability).

- I would strongly recommend to additionally show mean values, standard deviations, and (bootstrapped) confidence intervals together with other important statistical values (e.g., z-values). Presenting only p-values (e.g., Table 3) is not enough.

Author Response

(The authors gave the same response as above.)

Round 2

Reviewer 2 Report

Authors have addressed all of my comments.